# Identification of Somatic Mutations in Plasma Cell-Free DNA from Patients with Metastatic Oral Squamous Cell Carcinoma

**DOI:** 10.3390/ijms241210408

**Published:** 2023-06-20

**Authors:** Li-Han Lin, Kuo-Wei Chang, Hui-Wen Cheng, Chung-Ji Liu

**Affiliations:** 1Department of Medical Research, MacKay Memorial Hospital No. 92, Sec. 2, Chung San N. Rd., Taipei 10449, Taiwan; 2Institute of Oral Biology, School of Dentistry, National Yang Ming Chiao Tung University, Taipei 11221, Taiwan; 3Department of Stomatology, Taipei Veterans General Hospital, Taipei 11121, Taiwan; 4Department of Oral and Maxillofacial Surgery, Taipei MacKay Memorial Hospital, Taipei 10449, Taiwan

**Keywords:** cell-free DNA, distant metastasis, liquid biopsy, mutation burden, oral cancer, whole-exome sequencing

## Abstract

The accurate diagnosis and treatment of oral squamous cell carcinoma (OSCC) requires an understanding of its genomic alterations. Liquid biopsies, especially cell-free DNA (cfDNA) analysis, are a minimally invasive technique used for genomic profiling. We conducted comprehensive whole-exome sequencing (WES) of 50 paired OSCC cell-free plasma with whole blood samples using multiple mutation calling pipelines and filtering criteria. Integrative Genomics Viewer (IGV) was used to validate somatic mutations. Mutation burden and mutant genes were correlated to clinico-pathological parameters. The plasma mutation burden of cfDNA was significantly associated with clinical staging and distant metastasis status. The genes *TTN*, *PLEC*, *SYNE1*, and *USH2A* were most frequently mutated in OSCC, and known driver genes, including *KMT2D*, *LRP1B*, *TRRAP,* and *FLNA*, were also significantly and frequently mutated. Additionally, the novel mutated genes *CCDC168*, *HMCN2, STARD9*, and *CRAMP1* were significantly and frequently present in patients with OSCC. The mutated genes most frequently found in patients with metastatic OSCC were *RORC*, *SLC49A3*, and *NUMBL.* Further analysis revealed that branched-chain amino acid (BCAA) catabolism, extracellular matrix–receptor interaction, and the hypoxia-related pathway were associated with OSCC prognosis. Choline metabolism in cancer, O-glycan biosynthesis, and protein processing in the endoplasmic reticulum pathway were associated with distant metastatic status. About 20% of tumors carried at least one aberrant event in BCAA catabolism signaling that could possibly be targeted by an approved therapeutic agent. We identified molecular-level OSCC that were correlated with etiology and prognosis while defining the landscape of major altered events of the OSCC plasma genome. These findings will be useful in the design of clinical trials for targeted therapies and the stratification of patients with OSCC according to therapeutic efficacy.

## 1. Introduction

Oral squamous cell carcinoma (OSCC) is a relatively common malignancy of the upper aerodigestive tract with poor prognosis and a high mortality rate. In Asia, 248,360 new OSCC cases (ICD10 C00-06—Lip, oral cavity) were diagnosed in 2020, among which 131,610 involved the death of the patient from the disease [1]. In about 4.0–7.4% of patients with OSCC, several tumors were found to be developing simultaneously in the head and neck region [2,3]. In addition to the high incidence and mortality rates worldwide [4], OSCC has a high recurrence rate after treatment, which may be because multiple lesions develop concurrently and over a large mucosal area [5]. To ensure the appropriate diagnosis and treatment of OSCC, the underlying genomic changes associated with the carcinogenesis of the disease must be understood.

Cell-free DNA (cfDNA) can originate from normal cells, including normal leukocytes undergoing apoptosis, or may be shed from dead normal cells or cancer cells. Thus, cfDNA is detectable in all patients, regardless of their health or cancer status [6,7,8]. Indeed, total cfDNA may consist of normal and tumor cfDNA in variable proportions [8]. For the diagnosis and surveillance of cancer, there is growing interest in plasma cfDNA, i.e., “liquid biopsy” [9,10,11,12]. cfDNA may originate from circulating tumor cells; therefore, the levels of cfDNA can reflect the presence of micrometastatic disease and the aggressiveness of the cancer [13]. Higher plasma cfDNA levels have been found in patients with oropharyngeal squamous cell carcinoma (OPSCC), a subtype of head and neck cancer, than in patients with other types of head and squamous cell carcinoma (HNSCC) [14]. For example, Lin et al. found 2.2-fold higher concentrations of cfDNA in patients with OSCC than in healthy controls [15].

Whole-exome sequencing (WES) has provided new insights into the molecular basis of OSCC progression [16,17,18]. Thus, WES analyses of cfDNA could lead to the discovery of novel resistance mechanisms, characterization of mutational signatures predicting treatment responses, identification of genes associated with disease progression, and/or capture of DNA signatures that aid treatment-associated decision-making [19,20,21]. To improve the diagnosis of individuals at risk and the treatment of OSCC, more sensitive and specific biomarkers are required [22,23]. Current evidence suggests that assessing cfDNA-derived somatic mutations in the plasma of OSCC patients according to a panel of genes could serve as a diagnostic or recurrence monitoring test [24,25,26,27,28,29,30].

Liebs et al. published the cfDNA mutation profiles of six metastasized HNSCC patients [30]. However, cfDNA has not been analyzed to determine the whole-exome somatic mutations in primary OSCC. Therefore, the aim of the present study was to obtain WES data from oral cancer cfDNA by sequencing 50 OSCC paired plasma/blood samples using a whole-exome platform. Somatic mutations were defined by referring to high-quality data obtained previously and using independent data and variant calling methods. The selected data were then combined and validated using the Integrative Genomics Viewer (IGV) to reduce false-positive calls but maintain the mutation detection sensitivity of the method. Finally, mutations identified as truly somatic were compared with clinical data.

## 2. Results

### 2.1. Characterization of Patients

In the paired analysis of the cfDNA and germline DNA of 50 OSCC patients, the fragment size distribution of cfDNA was ~150–200 bp (Appendix A), and cfDNA concentrations of 20–473 ng/mL were distributed in a wide range (Appendix A). Table 1 shows the clinical characteristics of the study subjects: 48 males and 2 females aged 40–89 years (mean age: 59.6 years). In Taiwan, the OSCC was male predominated and the male to female ratio was 16:1. Their primary lesions were most commonly located in the buccal region and the gingiva (28% and 24%, respectively).

### 2.2. cfDNA WES

WES data (cfDNA and matched normal samples) were obtained for all patients with a mean coverage depth of 90.87× and high coverage uniformity (98.18% of amplicons covered at 10× mean coverage). The WES data analysis flowchart is shown in Figure 1. The tools Muse, Mutect2, SomaticSniper, Sterlka2, and VarDict were used for variant calling to identify somatic mutations in the 50 paired cfDNA/normal samples and found 28,876, 1993, 21,763, 45,100, and 44,506 somatic mutations, respectively, with 84,365 somatic mutations found in total in the target (coding and splicing) region (Figure 1).

### 2.3. Validation and Assessment of Somatic Mutation Calling

IGV was used to examine somatic mutations in the frequently mutated genes according to the previous study [18,31,32]. The IGV screenshot shows somatic mutations that passed or did not pass the IGV examination (Appendix A). IGV-passed mutations were represented in the Appendix A. Base calls with ≥2 mismatches with in a 20 bp window were considered false-positive mutations (Appendix A). As shown in Table 2, the most frequently observed non-PASS genes during IGV examination were *MUC19* (49.1%), *OBSCN* (38.5%), *KMT2D* (35.5%), *RYR1* (23.5%), and *MUC16* (22.5%). The high mutation levels of genes were correlated with high false-positive variants (Spearman’s rho = 0.763, *p* < 0.001). Moreover, the loci of *MUC16* and *MUC19* had numerous recurrent false-positive variants in both cfDNA and normal samples (Appendix A). Therefore, we filtered out the *MUC16* and *MUC19* mutations to reduce false-positive calls. After filtering, 84,045 somatic mutations remained in 14,733 unique genes, including 22,394 synonymous mutations, 54,310 missense mutations, 1136 splicing site mutations, 7 frameshift mutations, 22 in-frame mutations, 97 lost starts, 5982 inserted stops, 90 stop loss mutations, and 7 retained stops.

### 2.4. Correlations between Mutation Burden and Sequencing Quality and Clinical *Parameters*

Plasma cfDNA mutation burden (PMB) was calculated as the number of somatic mutations at a VAF of ≥5% in the coding region per Mb. The mean PMB was 18.46 mutations per Mb per patient. As depicted in Appendix A, PMB was a low/moderate correlation between cfDNA concentration (Spearman’s rho = 0.328, *p* = 0.020) and 0.2× mean coverage (Spearman’s rho = (–)0.320, *p* = 0.023). Although PMB was negatively correlated with coverage uniformity, all WES data showed better coverage uniformity (mean of 0.2× mean coverage: 98.18 ± 0.97%; range: 93–99%; Appendix A). A high PMB was strongly associated with distant metastasis (*p* = 0.002) and an advanced clinical stage (*p* = 0.019) (Table 1 and Figure 2). However, only the difference in distant metastasis was significant after Bonferroni correction. After adjusting for the effect of age, distant metastasis was also associated with PMB in Appendix A (B = 0.320, *p* = 0.025).

We further investigated the relationship between different mutation types and clinical parameters. As shown in Table 3, higher number of missense and splicing site mutations were significantly associated with advanced clinical stage (*p* = 0.013 and *p* = 0.006, respectively). However, none of these differences remained significant after Bonferroni correction. Higher number of synonymous, missense, inserted stop, splicing site, stop loss and loss start mutation were significantly associated with advanced clinical stage (*p* = 0.003, *p* = 0.002, *p* = 0.003, *p* = 0.001, *p* = 0.023 and *p* = 0.001, respectively). However, only the difference in synonymous, missense, inserted stop, splicing site, and loss start mutation were significant after Bonferroni correction.

### 2.5. Analysis of Plasma cfDNA Revealed Clinically Actionable Mutations without Prior Knowledge of the Tumor

Analysis of the cfDNA WES data indicated that the most frequently mutated genes were *TTN* (48%), *PLEC* (46%), *SYNE1* (44%), and *RYR3* (44%) (Figure 3). To predict the potential mutational driver genes in OSCC, the InToGene platform and the datasets of Bailey et al. were used [33]. Four cancer driver genes, namely *KMT2D*, *LRP1B*, *TRRAP,* and *FLNA*, were identified in the 20 most frequently mutated gene sets. In the TCGA-OSCC dataset, the most frequently mutated genes having been observed were *TP53* (68%), *TTN* (42%), *FAT1* (26%), and *CDKN2A* (22%) (Appendix A). The mutational landscape in our cfDNA appeared dissimilar to that in the TCGA-OSCC dataset (Appendix A). In addition, we compared the TCGA-OSCC mutation profiles with our data. The novel mutated genes *CCDC168*, *HMCN2, STARD9*, and *CRAMP1* were significantly and frequently present in patients with OSCC. The mutation frequencies of *CCDC168*, *HMCN2, STARD9*, and *CRAMP1* were 34%, 32%, 32%, and 30%, respectively.

The relationship between nonsynonymous mutations and clinical parameters was also examined in the 20 most frequently mutated genes. The presence of metastasis in patients with OSCC was significantly associated with nonsynonymous mutations in *TTN*, *SYNE1*, *RYR3*, *DMD*, *HECTD4*, *KMT2D*, *NEB*, *DNAH10*, *SYNE2*, *VPS13D*, *ZFHX4*, and *LRP1B* (Appendix A). However, only the difference in *KMT2D* and *DNAH10* were significant after Bonferroni correction. Kaplan–Meier survival curve analysis revealed that the mutation of *KMT2D* tended toward an association with a poor outcome, but the association was not significant (log rank *p* = 0.090). Moreover, no significant association was found between the gene mutations and other clinical parameters. Several of these genes are known to be involved in tumorigenesis, including *SYNE1, RYR3, HECTD4, KMT2D,* and *LRP1B* [34,35,36,37,38]. Furthermore, mutations in *TTN*, *KMT2D*, *ZFHX4*, and *LRP1B* are associated with poor outcomes [39,40,41].

The 300 most frequently mutated gene sets in the cfDNA WES data were used to conduct protein–protein interaction (PPI) network analysis and identify hub genes (Appendix A). *KMT2A*, *CREBBP*, *MTOR,* and *ITPR1* were identified as hub genes with the highest predicted probability of controlling different gene clusters in cfDNA (Appendix A). These genes are involved in the regulation of chromatin-mediated transcription (*KMT2A*, *KMT2C*, and *KMT2D),* fundamental signal transduction (*MTOR*), and calcium signaling (*ITPR1*, *ITPR2*, and *ITPR3*). In addition, the dysregulation or mutation of these genes in cancer cells affects cell growth, survival, metabolism, and metastasis [41,42,43,44].

### 2.6. Distant Metastasis and Survival-Related Genes

We noted the significantly altered mutation frequency of genes between patients with distant metastatic OSCC and nonmetastatic OSCC (Figure 4A). *UMOD* was the most frequently mutated gene in patients with nonmetastatic OSCC, whereas *RORC*, *SLC49A3*, and *NUMBL* were the most frequently mutated genes in patients with metastatic OSCC. Deregulation expression of NUMBL and RORC have been reported to be involved in the regulation of cancer cell migration, invasion, and metastasis [45,46]. However, the role of *SLC49A3* in cancer is unclear. To identify the genetic mechanisms associated with metastasis, the 200 most frequently mutated gene sets in patients with metastatic OSCC were used to conduct PPI network analysis, which revealed that *NCOA1* and *CBL* were regulators with the highest predicted probability of controlling different gene clusters (Appendix A). *NCOA1* and *CBL* have previously been shown to play oncogenic roles in many cancers [47,48].

The significantly altered mutation frequency of genes between surviving and expired patients is shown in Figure 4B. The most frequently mutated gene in patients that survived OSCC was *STAB1*, whereas the most frequently mutated genes in expired patients were *EIF4G1*, *PLOD3*, and *FAM208A*. *STAB1*, *EIF4G1*, and *PLOD3* have been reported to regulate the proliferation, migration, and invasion of cancer cells [49,50,51]; however, the role of *FAM208A* in cancer is unclear. A PPI network of proteins encoded by the 300 most frequently mutated gene sets in expired patients was constructed, and four hubs (*VCP, PPP2R1A, EHHADH,* and *ACAT1*) were identified (Appendix A). Mutations of *PPP2R1A* in uterine cancer affect oncogenic signaling and promote tumor cell growth [52], whereas *EHHADH* and *ACAT1* are regulators of drug resistance in tumor cells [53,54]; however, the role of *VCP* in cancer is unclear.

### 2.7. Molecular Pathway Analysis

The 300 most frequently mutated genes in all patients, patients with metastatic OSCC, and expired patients were imported into Annotation, Visualization and Integrated Discovery (DAVID) database, and Kyoto Encyclopedia of Genes and Genomes (KEGG) pathway enrichment analysis was performed. As shown in Table 4, the top three enriched KEGG pathways in all patients, namely extracellular matrix (ECM)-receptor interaction, the calcium signaling pathway, and the Notch signaling pathway, had a significant false discovery rate (*p* < 0.05). The top three enriched KEGG pathways in patients with metastatic OSCC were choline metabolism in cancer, O-glycan biosynthesis, and protein processing in the endoplasmic reticulum. The top three enriched KEGG pathways in expired patients were branched-chain amino acid (BCAA) catabolism, ECM-receptor interaction, and renal cell carcinoma. The genes in renal cell carcinoma are involved in the regulation of the hypoxia response; thus, these genes were considered a hypoxia-related gene set.

The Kaplan–Meier method was used to analyze the association between OS time and the mutations of genes in these KEGG pathways. Mutations in *UGGT2*, *HSPA4L*, *COL2A1*, *AUH*, *ACADSB*, *ARNT*, *EPAS1*, *PAK3*, and *TFE3* were associated with poor outcomes (Appendix A). However, only the difference in *HSPA4L* and *ACADSB* were significant after Bonferroni correction. Mutations in the BCAA catabolism gene set were detected in 12 (24%) patients (Figure 5A). Mutations in the calcium signaling-related genes *HMGCS2*, *AUH*, *ACAT1*, *ACADSB*, and *EHHADH* were found in 6 (12%), 6 (12%), 6 (12%), 4 (8%), and 5 (10%) patients, respectively (Appendix A). When patients had mutations in the BCAA catabolism gene set, their prognosis was poorer than that of patients without such mutations. However, the difference in the BCAA catabolism gene set was not significant after Bonferroni correction. Mutations in the hypoxia-related gene set were found in 15 (30%) patients (Figure 5B); the affected genes were *ARNT*, *EPAS1*, *PAK3*, *RAF1*, and *TFE3* in 8 (16%), 4 (8%), 5 (10%), 6 (12%), and 7 (14%) patients, respectively (Appendix A). When patients had mutations in the hypoxia-related gene set, their prognosis was poorer than that of patients without such mutations. The difference in the hypoxia-related gene set remained significant after Bonferroni correction. We also identified Food and Drug Administration (FDA) approved drugs associated with the candidate genes. Two candidate genes, *HADHA* and *HADHB*, which are involved in the regulation of the BCAA catabolism pathway (Appendix A), were found to be susceptible to treatment with the FDA-approved drug triheptanoin.

## 3. Discussion

When using commonly available WES technologies, large amounts of circulating DNA are needed, which cannot be obtained from adequate volumes of plasma samples. In WES based on hybridization, the amount of circulating DNA and the complexity of the sequencing library can limit the process since a higher number of PCR cycles is required when the input material is limited. To address this issue, we utilized the ThruPLEX-FD Prep Kit (Rubicon Genomics, Inc., Ann Arbor, MI, USA) to maximize the yield for library generation [55,56,57]. cfDNA is usually present at low concentrations and highly fragmented, and its abundance depends on the cancer type and stage, as well as the sample treatment prior to analysis [58,59]. Shearing and other fragmentation methods have a considerable effect on the size distribution within fragments and therefore the results of analysis. Given the complexity of the complete workflow, including the preparation of samples and libraries, sequencing, and data analysis, the process should be standardized to ensure that data quality is optimized, especially when clinical cohorts are investigated [60]. The ThruPLEX system has high sensitivity, making possible the detection of more low-abundance somatic mutations than can be detected with QIAseq (Qiagen, Hilden, Germany), NEXTFLEX (BioScientific, Austin, TX, USA), Accel (Swift Biosciences, Ann Arbor, MI, USA) with PCR, and Accel PCR-freer kits [60]. Moreover, the ThruPLEX kit also allows the analysis of variants from various types of plasma samples [60]. In the present study, we combined library preparation via ThruPLEX-FD with exome enrichment via SureSelect technology to achieve 98.18% coverage and 90.87-fold depth. Our evaluation of cfDNA sequencing data demonstrates the high performance of the established workflow combining ThruPLEX-FD library preparation with SureSelect technology for exome enrichment. To broaden the detection range of mutations, we employed the NGS technology platform (Illumina, San Diego, CA, USA) with high depth of sequences of the whole exon region. We obtained better mapping performance, enrichment efficiency, target coverage, and sequencing depth between cfDNA reads than another study with a depth of 49× (minimum: 40×) and exome coverage of 82% [21]. Another study was performed with a median sequencing depth of 55.59-fold for cfDNA WES [61]. Nonetheless, potential sources of error, such as content bias depending on the library generation method and amplification errors, must be considered. Addressing these potential sources of error through assay design should be considered during the design phase of experiments [62].

Whole-genome sequencing and WES of cfDNA have the potential to detect genetic variation at the nucleotide level, which could enable identification of subclonal tumors [63]. However, not all mutations found in plasma cfDNA necessarily originate from tumors. For instance, Dietz et al. reported a limited correlation (∼5–57%; median: 17.2%) between somatic cfDNA mutations and matched tumor samples in non-small cell lung cancer patients [57]. In contrast, other studies have found 11–92% of tissue mutations in the plasma of patients with HNSCC [30,64,65]. The source of this variation could include non-sampled tumor populations, distant metastases, or normal cells unrelated to the tumor. While germline variability can be filtered out by comparing with individually matched germline DNA and using databases like dbSNP (as done in this study), the specificity of cfDNA genomics will remain a challenge until more is known about the genetic aberrations in normal tissue and their representation in cfDNA [10]. The observation that the opposite was observed in HNSCC patients requires validation in a larger patient cohort.

In this study, we demonstrated the utility of WES for identifying cfDNA variants in plasma samples from cancer patients. Somatic mutations, including SNPs and known annotated mutations from the COSMIC database, were identified and removed from the datasets [18,66]. Germline variability was eliminated by comparing the data with individually matched germline DNA and applying high-fidelity filtering against databases such as dbSNP, PoN panel, oxodG artifacts, and strand bias [18,66]. To identify true somatic mutations, it is useful to combine the results of multiple mutation callers to reduce false positives while maintaining analysis sensitivity [67]. However, false-positive mutations may still be present in the datasets, even when normal panel SNPs and repeat-rich sequences have been removed [66]. IGV validation revealed false positives in some genes, particularly in gene variants with high mutation levels. A combination of at least two callers provides better performance than a single caller alone, based on the number of true and false detections in any combination of mutation callers across all replicates [18,66,67].

In the present study, we explored the relevance of using liquid biopsy to characterize the mutational landscape of OSCC using WES with deep coverage. We detected numerous different novel mutations at low frequencies in pathways possibly associated with OSCC oncogenesis. These included ECM-receptor interaction [68], arginine and proline metabolism [69], deregulated choline metabolism [70], and branched-chain amino acid metabolism [71]. Concordance between the variants determined by sequencing tumor tissue and cfDNA was low in previous studies [30,65]. Recurrent hotspot mutations in specific genes identified using a highly sensitive amplification-based method weas shown to have a higher sensitivity in OSCC [28]. Although the complete molecular landscape and tumor heterogeneity of OSCC are not yet fully understood, WES gene panels, such as the one used in this study, can provide a more comprehensive dataset of mutated genes in samples, as opposed to targeted panels that include only a limited number of genes [29,72,73,74,75,76]. This approach has the potential to identify genes of interest related to treatment resistance and capture DNA signatures that can help make treatment-related decisions without prior knowledge of the mutational profile [19,61].

Our data were discriminative from the TCGA database. In metastatic colon rectum cancer, gastric cancer, and endometrial carcinoma, cfDNA gene mutation patterns were quite different when compared with the TCGA dataset [77,78,79]. This may results from the difference of sample patterns for patients enrolled in the studies. Asian patient cases are underrepresented in the TCGA, which are almost entirely non-metastatic surgical tissue samples. This may cause the discrepancy from the cfDNA scheme compared with the TCGA molecular classification scheme.

The level of cfDNA in the blood circulation can be affected by various factors, including tumor size, stage, and growth rate. The broader and deeper the cfDNA assay employed, the higher the likelihood of detecting mutations [20]. In addition, the total number of tumor-derived mutations interrogated is another important factor affecting the detection of mutation burdens in plasma. Higher levels of cfDNA in the blood circulation have been observed primarily in patients with a tumoral mass rather than patients without tumors [15]. In the case of OSCC, plasma cfDNA levels were significantly higher in patients with larger tumors, cervical lymph node metastasis, and late-stage cancer. These levels were also positively correlated with poor prognosis [15]. Thus, the level of cfDNA reflects tumor progression to some extent. In the present study, PMB was correlated with plasma concentration. The levels of cfDNA were significantly higher in patients with a higher PMB, a phenomenon that was also observed in hepatocellular carcinoma [61]. The mutational burden of cfDNA has been related to tumor size, tumor growth rate, or cell turnover, as shown using a theoretical mathematical model of cfDNA shedding in lung cancer [80]. In our study, higher PMB was significantly associated with unfavorable clinical parameter such as distant metastasis, after Bonferroni correction. However, our statistical power was less than 0.8 (power = 0.61); therefore, we could not eliminate the possibility of a type 2 error. The power analysis revealed that the sample size of N = 78 was required to achieve 80% power.

In particular, high levels of PMB were found in patients with distant metastatic OSCC. Similar findings were reported for rectal cancer, in which presurgery cfDNA was used to detect patients with minimal metastatic disease and identify those at high risk of distant recurrence [81]. This issue should be investigated further by collecting and analyzing more samples.

Previous studies have shown that tumor-derived cfDNA can be detected in 73% of patients with metastatic breast and prostate cancer, but cfDNA sensitivity in the metastatic setting may vary based on the location of the disease [81,82,83]. The ability of HNSCC tumor cells to metastasize depends on their ability to detach from the basement membrane and associated ECM components [84]. However, obtaining tumor biopsies can be limited by the location and timing of the biopsy, differential release of cfDNA among lesions, and tumor heterogeneity, which may limit the detection of tumor mutations [85,86]. The current study demonstrates that WES can be performed in a significant proportion of patients with metastatic oral cancers, enabling comprehensive clonal analysis of cfDNA to track tumor evolution and identify mechanisms of metastasis. It may be possible to develop a predictive algorithm that accurately distinguishes between metastatic and nonmetastatic cancer and identifies molecular-level mutational tumor types. Oncologists must rapidly screen for an increasing number of disease-specific or tumor agnostic biomarkers of drug response, but inadequate tumor tissue for comprehensive tumor profiling may delay appropriate systemic therapy administration. Patients with metastatic OSCC, which is difficult to biopsy, may be at risk of not receiving the most effective targeted therapies or curative immunotherapies. Liquid biopsies may soon be used to detect recurrent disease and select patients with OSCC for screening of distant metastasis during follow-up.

The mutational profile of a tumor plays a crucial role in determining the success of various therapeutic approaches, and precise targeting of these alterations, in combination with modified treatment regimens to reduce therapy resistance, can save many patients from disease and associated death. WES can identify mutations not included in targeted panels or global genomic features for samples with a high cfDNA tumor fraction. A more focused, sensitive assay capable of detecting clinically actionable mutations in cfDNA samples with low tumor fraction could also be employed. We believe this strategy will be of widespread interest as cfDNA profiling may become the initial tumor sequencing assay for many OSCC patients, enabling rapid identification of actionable drug targets before therapy initiation [87]. However, the clinical feasibility of such approaches is limited by the low fraction of cfDNA derived from tumors and the high cost of WES. Moreover, while more sensitive but focused cfDNA platforms can detect clinically actionable mutations covered by the assay design, they may not be able to detect low-frequency mutations in genes where the functional consequences of mutation remain unclear. Therefore, testing new compounds to target these genes may be necessary to improve the prognosis of OSCC.

## 4. Materials and Methods

### 4.1. Participants and Data Collection

Fifty patients with OSCC were enrolled in our study, which was approved by the institutional review board of MacKay Memorial Hospital (approval numbers: 12MMHIS178 and 15MMHIS104), after the patients provided informed consent. Demographic data, i.e., age, sex, clinical stage, and final status (survived or expired), were obtained from the patients’ medical records. For clinical staging, the tumor, node, and metastasis TNM classification of the American Joint Committee on Cancer (AJCC 7th edition) were used [88]. Exclusion criteria for this study were receiving adjuvant chemotherapy or radiotherapy before surgery.

TCGA-HNSCC WES dataset was downloaded from the Genomic Data Commons portal (https://portal.gdc.cancer.gov/) (accessed on 25 December 2021). We included only 387 patients where the primary site was located at the tongue, lip, mouth floor, tonsil, gums, palate, or oropharynx. This dataset was called “TCGA-OSCC” dataset [18]. Mutations located in the introns, intergenic regions, and in untranslated regions (UTR) were excluded.

### 4.2. DNA Extraction

cfDNA was extracted from 2 mL of plasma (obtained before OSCC-associated surgery) using a QIAamp Circulating Nucleic Acid Kit (Qiagen), as described previously [15]. Purified cfDNA was then eluted in 25 μL of elution buffer from the kit. The QIAamp Nucleic Acid Kit was used for sample collection instead of simple EDTA-coated tubes and blood samples were centrifuged immediately to avoid DNA fragmentation, which may achieve the best preservation of circulating cfDNA. cfDNA was quantified using a TapeStation 2200 (Agilent Technologies, Santa Clara, CA, USA), equipped with a high sensitivity D1000 ScreenTape system. This system analyzes up to 96 samples per run and resolves 35–1000 bp fragments, and the assay is suitable for the accurate sizing and quantification of DNA fragments in high-throughput applications (Appendix A) [15]. Paired whole-blood samples were collected in EDTA Vacutainer tubes (Becton Dickinson, Franklin Lakes, NJ, USA). Peripheral blood mononuclear cell negative fractions were used for germline DNA. Germline DNA was extracted from buffy coat layer prepared from 10 mL of whole blood using a QIAamp DNA Blood Mini Kit (Qiagen) according to the manufacturer’s instructions. The quality and quantity of genomic DNA were assessed using the TapeStation 2200 and a NanoDrop 2000 Photometer (Thermo Fisher Scientific, Waltham, MA, USA).

### 4.3. Sequencing of Plasma cfDNA

For plasma samples, ~1 ng of cfDNA was used as the input for library preparation, and libraries were prepared using a ThruPLEX DNA-seq Library Prep Kit according to the manufacturer’s protocol. For blood samples, libraries were prepared with 200 ng of genomic DNA using a SureSelect Library Preparation Kit (Agilent Technologies) according to the manufacturer’s instructions [21,55]. For exome enrichment, SureSelect Human All Exon V6 + UTRs probe sets were used. The captured libraries were amplified with 14 cycles of PCR, and the quality and concentration of libraries were assessed using the TapeStation 2200 system (Appendix A). Libraries were indexed with barcodes to allow sample pooling for multiplexed exome capture and sequencing. An IlluminaNovaSeq 6000 DNA sequencer (Illumina) was used to conduct WES [18].

### 4.4. Sequencing Data Processing

Somatic mutations were called using multiple pipelines (Figure 1). Sequencing data were aligned to the human hg38 genome using Burrows-Wheeler Aligner (v0.7.15). The Sequence Alignment/Map (SAM) file format was converted to the Binary Alignment/Map (i.e., BAM) format with SAMtools (v1.3.1). Five programs were used to call somatic mutations: Muse (v1.0rc), Mutect2 (v4.1.0.0), Strelka2 (v2.9.10), SomaticSniper (v1.0.5.0), and VarDict (v1.8.3) [89,90,91,92]. Plasma cfDNA somatic mutations were called via comparisons with matched control germline DNA. After variant identification, the variant calling file data from the five programs were merged using sample IDs and positions in the genome (e.g., ID-chr1-123).

Data were further filtered using the criteria reported in our previous study [18]: (i) removing common polymorphisms (SNPs), a minor allele frequency in the 1000 Genomes Project or Genome Aggregation Database (gnomAD) of >1%; (ii) Panel of Normal (the normal panel was created with the GATK tool “CreateSomaticPanelOfNormals”); (iii) 8-oxoguanine artifacts (identified and filtered using the GATK tools “FilterByOrientationBias” and “CollectSequencingArtifactMetrics”, respectively); (iv) removal of multiallelic sites, clustered events, and strand bias (estimated and filtered using the GATK tools “GetPileupSummaries,” “FilterMutectCalls,” and “CalculateContamination”, respectively); (v) <4 mutant alleles in the cfDNA sample and ≥4 mutant alleles in normal cells [93]; and (vi) variant allele frequency (VAF) of <0.05 in cfDNA tumor samples. Steensma DP suggested that next-generation sequencing has a mutation limit of detection at VAFs ~2% [64,94]. The thresholds of VAF was defined as >2% in our study. The filtered mutations were considered somatic mutations.

Variant Effect Predictor (v106; Ensembl: https://asia.ensembl.org/Homo_sapiens/Tools/VEP) (accessed on 13 May 2021). was used to annotate the somatic mutations [95]. By searching the InToGene platform (https://www.intogen.org/search, accessed on 13 May 2021) and the Bailey et al. datasets [33,96], potential mutational driver genes in OSCC were identified and annotated. Germline DNA and cfDNA WES data described in the present study were submitted to the Short Reads Archive database under BioProject accession numbers PRJNA749133 (https://www.ncbi.nlm.nih.gov/bioproject/?term=PRJNA749133, accessed on 13 May 2021) and PRJNA759378 (https://www.ncbi.nlm.nih.gov/bioproject/?term=PRJNA759378, accessed on 13 May 2021), respectively.

The plasma cfDNA mutation burden (PMB) was calculated according to the number of somatic mutations with a VAF of ≥5% in the target region of SureSelect Human All Exon v6 + UTR [91.08 megabases (Mb)] and is expressed as mutations per Mb [97,98]. The target region of the BED file is available at the SureDesign website (https://earray.chem.agilent.com/suredesign/, accessed on 13 May 2021).

### 4.5. Validation of Mutations

IGV (v2.13.1) was used to validate somatic mutations [18,99,100]. The 200 most frequently mutated genes were selected, and IGV was used to determine false-positive rates. To be considered “true-positive,” mutations were required to fulfill the following criteria: (i) the allelic configurations of the mutation were multiallelic variants; (ii) both forward and reverse strands had at least one mutant allele; (iii) <2 mismatches occurred within a 20 bp window; and (iv) the number of Alt alleles was <3 in normal cells and ≥3 in tumors [95].

### 4.6. Pathway Analysis

The 300 candidate genes with the highest mutation rates in all, metastatic, or expired patients were imported into a PPI network produced using STRING (v11.5; https://string-db.org/, accessed on 13 May 2021).

DAVID (https://david.ncifcrf.gov/, accessed on 13 May 2021) was used to further analyze the annotation results of the candidate genes. KEGG pathway functional enrichment results were analyzed using DAVID with a false discovery rate of <0.05.The candidate genes were matched with FDA–approved drugs (https://www.fda.gov/, accessed on 13 May 2021) using the United States FDA Table of Pharmacogenomic Biomarkers in Drug Labels.

### 4.7. Statistical Analysis

Data are presented as the SD ± standard deviation. For statistical analysis, rank correlation was conducted using Spearman and Mann–Whitney U tests. The categorical data was conducted by chi-square or Fisher’s test (expected number less than 5). Linear regression models were used to assess the association between clinical stage and distant metastasis with PBM, adjusted for age. Overall survival (OS) was defined as the duration from the time of first diagnosis to death or the last date of follow-up. OS was compared between two groups using Kaplan–Meier analysis. *p* values of <0.05 were considered statistically significant. To adjust for multiple tests, the *p* value for significance was adjusted by Bonferroni correction.

## 5. Conclusions

Based on the findings of this study, an array of five mutation callers was utilized to identify mutations in cfDNA samples from OSCC patients. The presence of PMB mutations was found to be associated with advanced disease stage and distant metastasis. In addition, novel somatic mutations were detected in cfDNA samples from patients with metastatic OSCC. This information could potentially inform the use of precision therapy approaches tailored to the specific mutations identified in individual patients’ cfDNA samples.

## Figures and Tables

**Figure 1 ijms-24-10408-f001:**
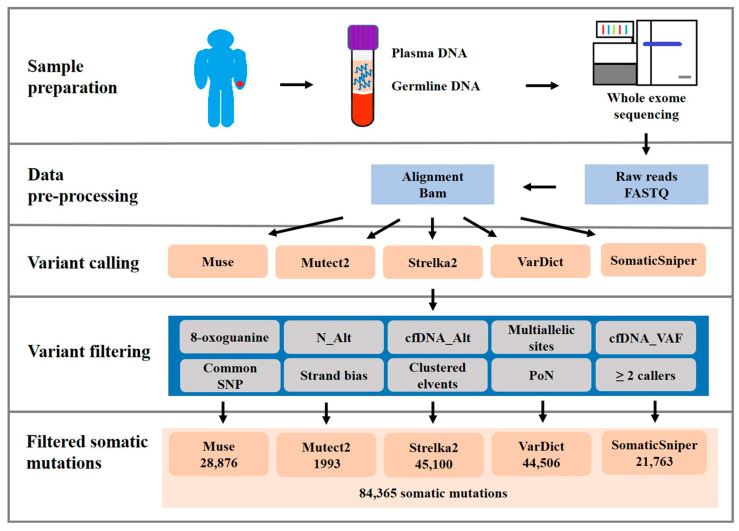
Flowchart illustrating the whole-exome sequencing (WES) analysis of cfDNA. WES of cfDNA was performed with matched normal samples from 50 OSCC patients. Somatic mutations were identified using five callers. The number of mutations called by each variant caller is depicted.

**Figure 2 ijms-24-10408-f002:**
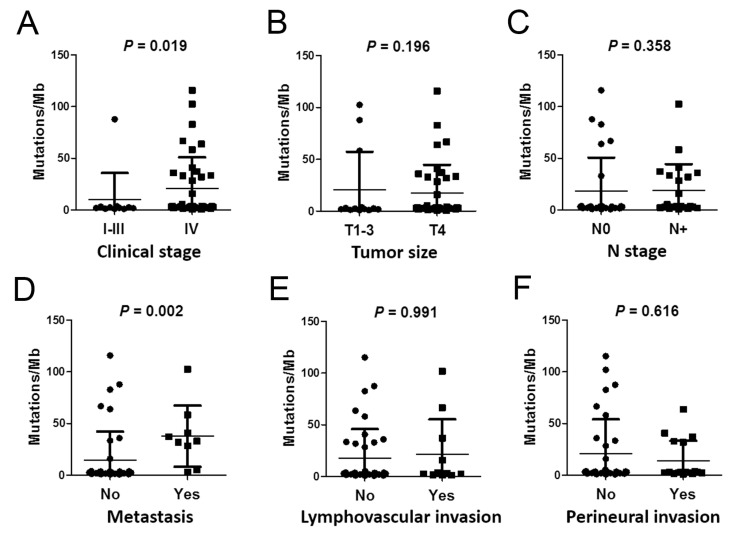
Scatter dot plot to illustrate the differential PMB associated with clinical parameters. Analysis of the PMB in OSCC patients according to (**A**) clinical stage, (**B**) tumor size, (**C**) N stage, (**D**) metastasis, (**E**) lymphovascular invasion, and (**F**) perineural invasion status. Each dot/square represent one sample value.

**Figure 3 ijms-24-10408-f003:**
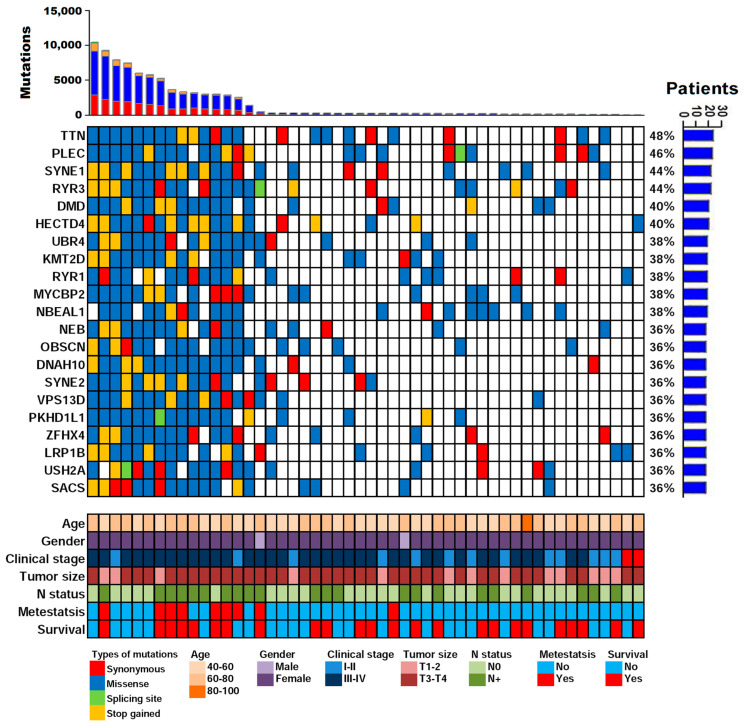
Mutation distribution of the most frequently mutated genes in cfDNA. Each column represents an individual OSCC patient, whereas each row denotes a mutated gene and clinical features. Clinical features and mutation types are color-coded as indicated. The panel on the right shows the number of mutations in the indicated gene.

**Figure 4 ijms-24-10408-f004:**
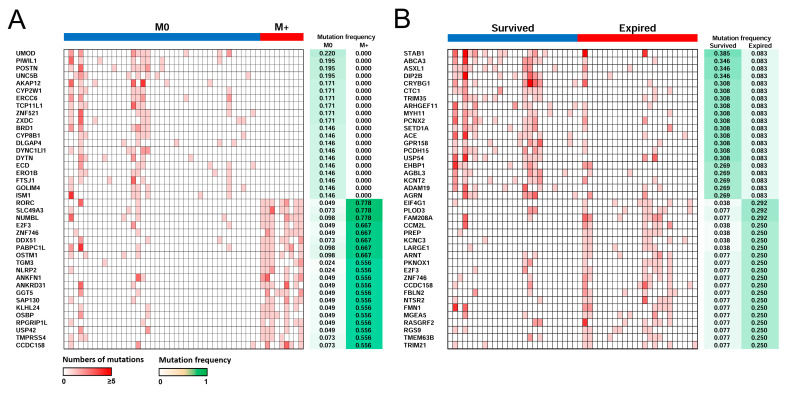
Heatmap of mutation profiles in cfDNA WES data according to metastasis and survival status. (**A**) The most frequently mutated genes in patients with metastatic and nonmetastatic OSCC are represented graphically. (**B**) Heatmap of the most frequently mutated genes in survived and expired patients. Panel on the right shows the frequency of mutations in the indicated gene.

**Figure 5 ijms-24-10408-f005:**
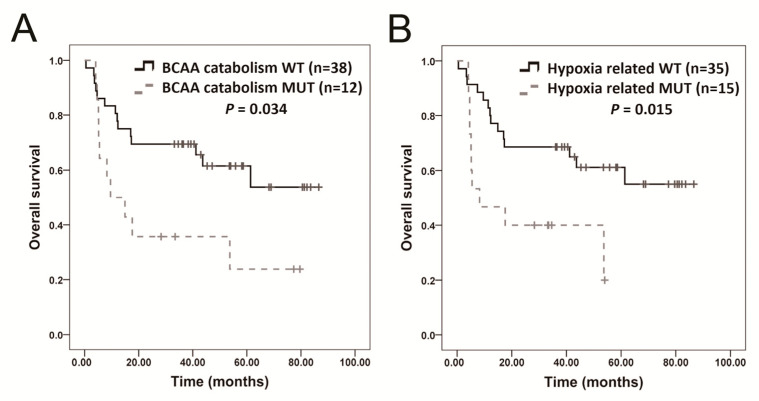
Mutations in branched-chain amino acid (BCAA) catabolism and hypoxia-related pathways are associated with patient survival. (**A**) Kaplan–Meier overall survival for patients with BCAA catabolism-mutated OSCC versus wild-type (WT) patients. (**B**) Kaplan–Meier overall survival for patients with hypoxia-related pathway-mutated OSCC versus WT OSCC. The Bonferroni-adjusted threshold for significance is set at α = 0.025 (0.05/2).

**Table 1 ijms-24-10408-t001:** Association between plasma cfDNA mutation burden (PMB) and clinical parameters.

Parameter	N	Mean of PMB ± SD		*p*-Value
Age				
≤60	23	16.41 ± 26.43		0.661
>60	27	20.19 ± 31.78		
Gender				
Male	48	19.05 ± 29.67		0.766
Female	2	4.13 ± 1.68		
T stage				
T1–3	13	20.68 ± 36.41		0.196
T4	37	17.67 ± 26.78		
N stage				
N0	28	18.03 ± 32.42		0.358
N+	22	19.0 ± 25.29		
Clinical stage				
I–III	11	9.89 ± 27.75		0.019
IV	39	20.87 ± 29.47		
Metastasis				
No	41	14.23 ± 33.22		0.002
Yes	9	37.7 ± 19.32		
Perineural invasion				
No	33	20.79 ± 28.25		0.616
Yes	17	13.92 ± 33.68		
Lymphovascular invasion				
No	39	17.56 ± 4.52		0.991
Yes	11	21.63 ± 10.15		
HPV status				
p16 negative	46	19.82 ± 30.08		0.391
p16 positive	4	2.67 ± 1.00		

Two groups were compared using Mann–Whitney U tests. The Bonferroni-adjusted threshold for significance is set at α = 0.006 (0.05/9).

**Table 2 ijms-24-10408-t002:** Detection of false-positive mutations in the most frequently mutated genes.

Gene	Total Mutations	IGV PASS		IGV Non-PASS	
		Mutations (%)		Mutations (%)	
*MUC19*	57	29	(50.9%)	28	(49.1%)
*OBSCN*	78	48	(61.5%)	30	(38.5%)
*KMT2D*	62	40	(64.5%)	22	(35.5%)
*RYR1*	51	39	(76.5%)	12	(23.5%)
*MUC16*	111	86	(77.5%)	25	(22.5%)
*DNAH1*	50	39	(78.0%)	11	(22.0%)
*PLEC*	51	40	(78.4%)	11	(21.6%)
*DNHD1*	39	31	(79.5%)	8	(20.5%)
*MACF1*	56	45	(80.4%)	11	(19.6%)
*USH2A*	37	30	(81.1%)	7	(18.9%)
*UBR4*	57	47	(82.5%)	10	(17.5%)
*FAT4*	35	29	(82.9%)	6	(17.1%)
*SYNE2*	53	44	(83.0%)	9	(17.0%)
*LRP1*	43	36	(83.7%)	7	(16.3%)
*LRP1B*	44	37	(84.1%)	7	(15.9%)
*HECTD4*	47	40	(85.1%)	7	(14.9%)
*INTS1*	30	26	(86.7%)	4	(13.3%)
*SYNE1*	87	78	(89.7%)	9	(10.3%)
*RYR3*	62	56	(90.3%)	6	(9.7%)
*TTN*	116	105	(90.5%)	11	(9.5%)
*DMD*	48	44	(91.7%)	4	(8.3%)
*ATM*	36	33	(91.7%)	3	(8.3%)
*DNAH11*	39	36	(92.3%)	3	(7.7%)
*NBEAL1*	26	24	(92.3%)	2	(7.7%)
*MYCBP2*	40	37	(92.5%)	3	(7.5%)

**Table 3 ijms-24-10408-t003:** Association between mutation types and clinical parameters.

	Clinical Stage			Metastasis		
Mutation Types	I–III	IV	*p* Value	No	Yes	*p* Value
Synonymous mutations	234.7 ± 174.7	508.0 ± 115.4	0.089	350.1 ± 105.0	893.3 ± 219.6	0.003
Missense mutations	572.9 ± 449.7	1231.0 ± 279.6	0.013	832.4 ± 249.0	2242.4 ± 594.1	0.002
Inserted stop mutations	80.5 ± 71.6	130.7 ± 37.9	0.083	92.3 ± 36.2	244.2 ± 74.4	0.003
Splicing site mutations	10.0 ± 8.6	26.3 ± 6.0	0.006	17.5 ± 5.5	46.3 ± 10.6	0.001
Stop loss mutations	1.0 ± 0.8	2.0 ± 0.8	0.264	1.7 ± 0.7	2.3 ± 0.9	0.023
Lost start mutations	1.5 ± 1.5	2.1 ± 0.6	0.055	1.6 ± 0.6	3.6 ± 0.8	0.001
Retained stop mutations	0.0 ± 0.0	0.2 ± 0.1	0.170	0.1 ± 0.1	0.1 ± 0.1	0.911
In-frame mutations	0.2 ± 0.1	0.5 ± 0.1	0.327	0.4 ± 0.1	0.8 ± 0.4	0.200
Frameshift mutations	0.0 ± 0.0	0.2 ± 0.1	0.170	0.1 ± 0.1	0.2 ± 0.1	0.324

The Bonferroni-adjusted threshold for significance is set at α = 0.006 (0.05/9).

**Table 4 ijms-24-10408-t004:** Most frequently mutated molecules in all patients, patients with metastatic OSCC, and expired patients.

KEGG Pathway	FDR *p*-Value	Molecules
All Patients		
ECM-receptor interaction	<0.001	FRAS1, COL4A4, COL4A6, COL6A3, COL6A5, HSPG2, LAMA2, LAMA3, TNXB, VWF
Calcium signaling pathway	0.002	ATP2B3, CACNA1B, CACNA1D, CACNA1G, ERBB4, ITPR1, ITPR2, PHKB, RYR1, RYR2, RYR3
Notch signaling pathway	0.017	CREBBP, NOTCH1, NOTCH2, NOTCH4, NCOR2
Patients with metastatic OSCC		
Choline metabolism in cancer	0.018	RAF1, SP1, WAS, DGKK, DGKQ, DGKZ, PIK3CB, SLC22A3, SLC44A1, SLC44A2
O-glycan biosynthesis	0.019	RFNG, ST6GAL1, GALNT14, GALNT18, GALNT5, GALNT8
Protein processing in endoplasmic reticulum	0.044	DNAJC10, NPLOC4, SEC23B, SEL1L, SEL1L2, STT3A, TRAF2, UGGT2, EIF2AK1, HSPA4L, P4HB, PDIA4, VCP
Expired patients		
BCAA catabolism	0.022	HMGCS2, AUH, ACAT1, ACADSB, EHHADH
ECM-receptor interaction	0.036	AGRN, COL2A1, COL9A1, ITGA9, ITGB5, LAMC1
Hypoxia-related	0.042	ARNT, EPAS1, PAK3, RAF1, TFE3

## Data Availability

All WES data described in the present study were submitted to the Short Reads Archive database (BioProject accession numbers: PRJNA749133 and PRJNA759378) and the SRA Run Selector project (https://www.ncbi.nlm.nih.gov/bioproject/?term=PRJNA749133 and https://www.ncbi.nlm.nih.gov/bioproject/?term=PRJNA759378).

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
