# Peer review of "Identification of Somatic Mutations in Plasma Cell-Free DNA from Patients with Metastatic Oral Squamous Cell Carcinoma"

_ijms, 2023, doi:10.3390/ijms241210408_

Round 1
Reviewer 1 Report
Identification of somatic mutations in plasma cell-free DNA from patients with metastatic oral squamous cell carcinoma is a paper aimed to obtain whole-exome sequencing data from oral cancer cell-free DNA by sequencing 50 oral squamous cell carcinoma (OSCC) paired plasma/blood samples using a whole-exome platform. The Authors identified molecular-level OSCC that were correlated with etiology and prognosis while defining the landscape of major altered events of the OSCC plasma genome. These findings will be useful in the design of clinical trials for targeted therapies and the stratification of patients with OSCC according to therapeutic efficacy.
The study is interesting and well done. However I think that some modifications would be necessary and could improve the quality of the paper. Firstly, I would suggest to the Authors to carry out a post-hoc analysis in order to assess the power of the study to demonstrate any significant association with etiology and prognosis given the large number of genes investigated. Overall, if the study was underpowered to find associations, readers should be advised of the likelihood that the lack of correlation could possibly be due to lack of power.
Secondly, the Authors should better clarify why they decided to present the data using the mean and standard error of the mean (SEM). The SEM is used in inferential statistics to give an estimate of how the mean of the sample is related to the mean of the underlying population. As the SEM is always smaller than the SD, the unsuspecting reader may think that the variability within the sample is much smaller than it really is. The SD and the SEM give two very different types of information. Whereas the SD estimates the variability in the study sample, the SEM estimates the precision and uncertainty of how the study sample represents the underlying population. In other words, the SD tells us the distribution of individual data points around the mean, and the SEM informs us how precise our estimate of the mean is. It is therefore inappropriate and incorrect to present data only as the mean (SEM) [Br J Anaesth. 2003 Apr;90(4):514-6].
Author Response
Reviewer 1
The study is interesting and well done. However I think that some modifications would be necessary and could improve the quality of the paper. Firstly, I would suggest to the Authors to carry out a post-hoc analysis in order to assess the power of the study to demonstrate any significant association with etiology and prognosis given the large number of genes investigated. Overall, if the study was underpowered to find associations, readers should be advised of the likelihood that the lack of correlation could possibly be due to lack of power.
Ans: Thank you for the comments.
The Bonferroni correction was used to adjusted p values for multiple testing. We also perform power test and reviesed the sentence in discussion “In our study, higher PMB was significantly associated with unfavorable clinical parameter such as distant metastasis, after Bonferroni correction. However, our statistical power was less than 0.8 (power = 0.61); therefore, we could not eliminate the possibility of a type 2 error. The power analysis revealed that the sample size of N = 78 was required to achieve 80% power.”
Secondly, the Authors should better clarify why they decided to present the data using the mean and standard error of the mean (SEM). The SEM is used in inferential statistics to give an estimate of how the mean of the sample is related to the mean of the underlying population. As the SEM is always smaller than the SD, the unsuspecting reader may think that the variability within the sample is much smaller than it really is. The SD and the SEM give two very different types of information. Whereas the SD estimates the variability in the study sample, the SEM estimates the precision and uncertainty of how the study sample represents the underlying population. In other words, the SD tells us the distribution of individual data points around the mean, and the SEM informs us how precise our estimate of the mean is. It is therefore inappropriate and incorrect to present data only as the mean (SEM) [Br J Anaesth. 2003 Apr;90(4):514-6].
Ans: We revised the Figure 2 and Table 1. The data are presented as mean± SD.
Reviewer 2 Report
The submitted manuscript describes the clinical possibilities of whole exome sequencing of blood cell-free DNA (cfDNA) of oral squamous cell carcinoma patients (OSCC) for the prediction of cancer stage or metastasis and the prospect of new therapeutic targets. The topic is interesting and falls within the scope of International Journal of Molecular Sciences. I have suggestions on the present form of the manuscript.
1. Page 5, Table 1: The tumor mutation burden was associated with the patient’s age and clinical stage in Table 1. Was the effect of clonal hematopoiesis correctively excluded in this analysis? Or it is necessary to statistically exclude any age bias in clinical stage or metastatic status as mutations derived from clonal hematopoiesis might increase with aging.
2. Page 5, Table 1: It was obscure what the values of “Mean” meant in Table 1. I guess that the “tumor burden” in the table title means tumor mutation burden. If so, I recommend replacing “tumor burden” with “tumor mutation burden.” And the author should include “units” of the values in the table.
3. Page 14, lines 403–410: I understood as the authors excluded SNPs and known annotated mutations for detecting novel mutations of OSCC cfDNA. This exclusion method and criteria should be described in materials and methods.
Although the overall quality of English writing was fine, minor textual errors should be corrected.
Author Response
The submitted manuscript describes the clinical possibilities of whole exome sequencing of blood cell-free DNA (cfDNA) of oral squamous cell carcinoma patients (OSCC) for the prediction of cancer stage or metastasis and the prospect of new therapeutic targets. The topic is interesting and falls within the scope of International Journal of Molecular Sciences. I have suggestions on the present form of the manuscript.
- Page 5, Table 1: The tumor mutation burden was associated with the patient’s age and clinical stage in Table 1. Was the effect of clonal hematopoiesis correctively excluded in this analysis? Or it is necessary to statistically exclude any age bias in clinical stage or metastatic status as mutations derived from clonal hematopoiesis might increase with aging.
Ans: The Age adjusted multiple linear regression models were performed in supplementary Table S1
- Page 5, Table 1: It was obscure what the values of “Mean” meant in Table 1. I guess that the “tumor burden” in the table title means tumor mutation burden. If so, I recommend replacing “tumor burden” with “tumor mutation burden.” And the author should include “units” of the values in the table.
Ans: We revised the Table 1. We replaced “mean” with mean of cfDNA tumor burden.
- Page 14, lines 403–410: I understood as the authors excluded SNPs and known annotated mutations for detecting novel mutations of OSCC cfDNA. This exclusion method and criteria should be described in materials and methods.
Ans: We revised the sentence. “Data were further filtered using the criteria reported in our previous study (18): (i) removing common polymorphisms (SNPs), a minor allele frequency in the 1000 Genomes Project or Genome Aggregation Database (gnomAD) of >1%”.
Round 2
Reviewer 2 Report
The manuscript and supplementary data have been much improved. I think this manuscript will be acceptable.